# The Impact of Solid-Phase Fermentation on Flavonoids, Phenolic Acids, Tannins and Antioxidant Activity in *Chamerion angustifolium* (L.) Holub (Fireweed) Leaves

**DOI:** 10.3390/plants12020277

**Published:** 2023-01-06

**Authors:** Marius Lasinskas, Elvyra Jariene, Nijole Vaitkeviciene, Jurgita Kulaitiene, Aida Adamaviciene, Ewelina Hallmann

**Affiliations:** 1Department of Plant Biology and Food Sciences, Agriculture Academy, Vytautas Magnus University, Donelaicio St. 58, 44248 Kaunas, Lithuania; 2Department of Agroecosystems and Soil Sciences, Agriculture Academy, Vytautas Magnus University, Donelaicio St. 58, 44248 Kaunas, Lithuania; 3Department of Functional and Organic Food, Institute of Human Nutrition Sciences, Warsaw University of Life Sciences, Nowoursynowska 15c, 02-776 Warsaw, Poland; 4Bioeconomy Research Institute, Agriculture Academy, Vytautas Magnus University, Donelaicio St. 58, 44248 Kaunas, Lithuania

**Keywords:** polyphenols, solid-phase fermentation, fireweeds, antioxidant activity

## Abstract

At present, the consumption of medical plants and functional foods is growing across the whole world. Fireweed (*Chamerion angustifolium* (L.) Holub), an important medicinal plant that has various pharmacological effects (antioxidant, anti-inflammatory, anticancer and others), can improve the state of health and well-being and reduce the risk of various diseases. The aim of this work was to investigate polyphenols (flavonoids, phenolic acids and tannins) and antioxidant activity in fireweed leaves fermented for 24, 48 and 72 h in solid-phase fermentation under aerobic and anaerobic conditions. High-performance liquid chromatography (HPLC) for polyphenols and the spectrophotometric method based on quenching of stable colored radical (ABTS^•+^) for antioxidant activity determinations were used. The results showed that the highest amounts of total polyphenols, total flavonoids and tannin oenothein B in the dried matter were found after 72 h and the highest total phenolic acids after 48 h of anaerobic solid-phase fermentation. The highest antioxidant activity was found after 72 h of solid-phase fermentation under aerobic conditions.

## 1. Introduction

Currently, more and more people are interested in food and medicinal plants, and they choose well-known and scientifically studied plants for consumption. One of these plants is fireweed (*Chamerion angustifolium* (L.) Holub), which belongs to the family of *Onagraceae* and the genus *Epilobium* L., which includes about 200 species that are widely distributed in the world [1]. Fireweeds are quite often confused with great willowherbs, which also belong to the genus *Epilobium* L. [2]. However, fireweeds clearly differ from great willowherbs in flower structure, leaf arrangement and other characteristics [3,4].

The most commonly used fireweed synonyms are *Epilobium angustifolium* L., *Chamerion angustifolium* (L.) Holub and *Chamaenerion angustifolium* (L.) Scop. According to Holub [5], *Chamerion* should be considered the correct Latin name for this genus. *Chamerion angustifolium* is called “fireweed” in the US and “rosebay willowherb” in Britain. Other synonymous names for this plant are found in the literature: perennial fireweed, narrow-leaved fireweed, willow herb, flowering willow, French willow and others [6].

The fireweed is a perennial herb with long and branched rhizomes and stout stolons. The green stems are often pink in color and grow up to 2 m tall. The willow-like leaves are alternate and 3–20 cm long. Blooms from late June to early September have large pink inflorescences and light brown seeds (1.0–1.3 mm long) with grayish-white hairs up to 13 mm long. One plant can mature in a year and produce 76,000 wind-borne seeds [7].

These plants are widespread and occur in the temperate zones of North America and Eurasia from 25° to 70° North latitude. They grow both in the lowlands and mountains. Fireweeds grow in man-made open habitats such as: cleared or burned forests, roadsides and railway embankments. Also, these plants are found at the edges of forests and thickets, riversides, slopes and meadows. Fireweeds grow best in a sunny place, but they can also grow in the shade. They like nutrient-rich and moderately moist soil. Fireweeds adapt to the conditions of their growing area [7].

Fireweeds are rich in polyphenolic compounds, especially tannins (ellagitannins), flavonoids and phenolic acids [8]. The healing properties of fireweed extracts are associated with the synergistic effect of polyphenols and ellagitannins (ellagitannin-oenothein B is the most abundant in fireweeds). Oenotein B is characterized by various pharmacological activities: antiandrogenic, antiproliferative, anticancer, antioxidant, anti-inflammatory and immunomodulatory [9].

The evaluation report of the European Medicines Agency states that the main indication for the raw material of the ground part of the fireweeds is to relieve the symptoms of diseases of the lower urinary tract associated with benign prostatic hyperplasia (BPH) [10]. In addition, fireweeds have antimicrobial [11], analgesic and antiaging properties [12].

*Chamerion angustifolium* has been ethnobotanically used as a medicinal (very strong antioxidant activity), edible, woody and ornamental plant. This plant of all the species of the genus *Epilobium* is the most well-known and often used in both folk and modern medicine around the world. Scientists are looking for ways to use fireweeds in the food and pharmaceutical industries. Therefore, it is very important to study and systematize knowledge about the chemical composition and pharmacological properties of the raw material and products of fireweeds. Fireweed leaves are usually collected at the beginning of July in the stage of the full flowering of plants when the plant synthesizes the most biologically active substances [13].

The phenolic acids and flavonoids in fireweeds have strong antioxidant properties [14]. Water-soluble extracts of the leaves of fireweeds have shown significant antioxidant activity in vitro studies. The antioxidant properties of different plant parts (roots, leaves and stems) were positively evaluated by Stajner et al. [15]. The highest antioxidant activity was found in the leaves of the fireweeds. The strong antioxidant effect was attributed to the high content of ellagitannins [16] and especially oenothein B [17]. Evaluation of the radical-scavenging activity of fireweeds collected during the growing season in relation to their flavonoid content was studied by Maruška et al. [18]. The highest content of flavonoids (8.71–11.12 mg 100 g^−1^ d.m.) and radical-scavenging activity were determined in fireweeds collected during the full flowering stage [18].

The biotechnological fermentation process is used in the food industry to change various organoleptic, physical and chemical properties of plant materials: smell, taste, color and better extraction of substances [19]. Recently, scientific interest in fermented foods has increased, leading to the development of various functional foods with some beneficial effects on human health [20]. The main advantage of such health products is that they are mild in action and very difficult to overdose, so they are reasonably safe [21]. Increasingly, people are consuming functional beverages—fermented herbal teas that have pleasant organoleptic properties and health effects [22]. Many people around the world consume green or black tea [23], but other fermented teas made from fireweed leaves are gaining interest. This tea tastes similar to green and black teas and has a long tradition of consumption [24].

One of the methods of making functional fireweed tea and improving the quantitative and qualitative composition of fireweed leaves and other products is the use of solid-phase fermentation technology. During this process, not only biochemical reactions take place inside the cells, but also the strong activity of microorganisms and enzymes [25]. Enzymes, produced during the metabolism of microorganisms (lactic acid bacteria and yeast) and solid-phase fermentation, such as polyphenol oxidase, etc., break down the macromolecular compounds (proteins, lipids and polysaccharides) contained in the leaves of the fireweeds into lower molecular weight substances and secondary metabolites products [26].

In addition, the crushing and pressing of the leaves during solid-phase fermentation can enhance the degradation processes of the cell walls and thus improve the diffusion of biologically active substances from the inner parts of the cells, which leads to more efficient extraction of compounds. It is believed that this may have been one of the main factors in determining higher polyphenol levels after the solid-phase fermentation process.

During solid-phase fermentation, no additional water is used, and the duration of fermentation is usually about 1–3 days, so this method of fermentation is more economically beneficial. In addition, solid-phase fermentation helps to avoid possible by-products of fungal metabolism, such as mycotoxins, resulting from their intense and prolonged activity in humid conditions, which are characteristic of conventional fermentation in a liquid medium.

A lot of research is carried out with the raw material of fireweeds in the world, but how the technological parameters of solid-phase fermentation (duration, aerobic or anaerobic environment) determine the qualitative and quantitative composition of biologically active substances are few. Based on these research results, we will be able to recommend which solid-phase fermentation conditions most significantly determine the increase of biologically active substances, and this will help manufacturers to produce high-quality food and pharmaceutical products from fireweed leaves.

## 2. Results

A very important feature of plants is resistance to adverse environmental conditions. This helps the plant to adapt and survive during a period when temperatures, humidity or other conditions are hindering the normal growth and development of the plant. The synthesis of various secondary metabolites (phenolic acids, flavonoids, tannins and others) is activated in response to climatic, humidity or other stressful effects on the plant. In this experiment, the vegetation of fireweeds in the year 2021 of this study was favorable for the growth of fireweeds and thus may have led to lower levels of secondary metabolites compared to other years [25].

### 2.1. The Contents of Polyphenols

The total biologically active substance quantities in the leaves of fireweeds after different solid-phase fermentation technological parameters in 2021 are presented in Table 1.

The presented data (Table 1) show that the amounts of total polyphenols did not change significantly after 24 and 48 h of aerobic solid-phase fermentation, but it increased by 9.26% after 72 h compared with the control variant. After 24 h of anaerobic solid-phase fermentation, the content of total polyphenols significantly decreased by 30.63%, but after 48 and 72 h, it increased by 10.48% and 13.94%, respectively, in comparison with non-fermented leaves.

The content of total phenolic acids significantly did not change after 24 h of aerobic solid-phase fermentation, but after 48 and 72 h decreased, respectively: 25.30% and 13.67%, compared with non-fermented fireweed leaves. After 24 h of anaerobic solid-phase fermentation, the content of total phenolic acids significantly decreased by 28.97%, and after 48 h, it increased by 17.28%, but further fermentation decreased the level of total phenolic acids by 12.37% compared with non-fermented fireweed leaves.

The amounts of total flavonoids decreased after 24 and 48 h of aerobic solid-phase fermentation (7.06% and 4.60%, respectively), but after 72 h did not change compared with the control variant. After 24 h of the anaerobic fermentation process, the content of total flavonoids significantly decreased by 46.52%, did not change after 48 h, but significantly increased by 81.26% after 72 h compared with non-fermented fireweed leaves.

The content of oenothein B in fireweed leaves increased in all aerobic solid-phase fermentation variants (11.77% after 24 h, 38.25% after 48 h and 45.18% after 72 h) compared with the control variant. After 24 h of the anaerobic fermentation process, the content of oenothein B decreased by 32.06%, and after 48 h did not significantly change, but after a longer fermentation process (72 h), it significantly increased by 49.68% compared with non-fermented fireweed leaves (Table 1).

### 2.2. The Contents of Phenolic Acids

Phenolic acids are secondary plant metabolites widely used in the treatment of various diseases caused by negative oxidative processes in the body (cancer, heart disease, etc.) [27]. Changes in phenolic acids‘ contents during solid-phase fermentation are shown in Table 2.

The results showed that, after 24 h of aerobic solid-phase fermentation, the ellagic acid content was 12.99% significantly lower compared with non-fermented fireweed leaves (control). After 48 h and 72 h of aerobic solid-phase fermentation, the amounts of ellagic acid were, respectively: 37.30% and 23.98%, significantly lower compared with leaves in the control variant. The ellagic acid amounts after the anaerobic solid-phase fermentation process were 33.57% significantly lower after 24 h and 23.88% lower after 72 h compared with non-fermented fireweed leaves. The highest content of ellagic acid was after 48 h of solid-phase fermentation under anaerobic conditions (Table 2).

*P*-coumaric acid contents after 24, 48 and 72 h of solid-phase fermentation under aerobic conditions decreased (by 5.47%, 7.56% and 1.22%, respectively) compared with non-fermented fireweed leaves. After anaerobic 24 h of solid-phase fermentation, the *p*-coumaric acid content decreased by 26.98%, but after 48 h and 72 h significantly increased, by 9.49% and 19.61%, respectively, compared with non-fermented fireweed leaves.

The gallic acid content in all fermented variants significantly increased compared with non-fermented fireweed leaves. The highest amount of gallic acid was detected after 24 h of aerobic solid-phase fermentation. After 48 h and 72 h of aerobic solid-phase fermentation, gallic acid amounts increased by 128.45% and 54.80%, respectively. During the anaerobic solid-phase fermentation process, the gallic acid content also increased by 30.75% after 24 h, by 212.77% after 48 h, and by 51.75% after 72 h compared with the control variant.

The highest amount of chlorogenic acid was in non-fermented fireweed leaves. After 24 h and 48 h of solid-phase fermentation under aerobic conditions, chlorogenic acid amounts significantly decreased, respectively: 51.05% and 54.27%, compared with the control variant. But after 72 h of the aerobic fermentation process, chlorogenic acid content did not change significantly, compared with non-fermented fireweed leaves. After anaerobic solid-phase fermentation, the chlorogenic acid amounts significantly decreased by 32.24% after 24 h, by 32.75% after 48 h and by 55.62% after 72 h compared with non-fermented fireweed leaves.

The benzoic acid content after 24 h of aerobic solid-phase fermentation significantly increased by 27.87%, and after 48 h decreased by 41.91%, but after 72 h did not change significantly compared with the control variant. After the anaerobic solid-phase fermentation process, the amounts of benzoic acid significantly increased by 22.83% after 24 h but decreased after 48 h and 72 h, respectively, by 54.66% and 13.08%, compared with non-fermented fireweed leaves (Table 2).

### 2.3. The Contents of Flavonoids

Flavonoids have been identified as a primary group of naturally occurring phenolic compounds in plants [28]. The flavonoid quantities in the leaves of fireweeds after different solid-phase fermentation technological parameters in the 2021 year are presented in Table 3.

The obtained data showed that the amounts of quercetin-3-*O*-rutinoside increased by about 8.53% after aerobic solid-phase fermentation compared with non-fermented fireweed leaves. After anaerobic solid-phase fermentation, the content of quercetin-3-*O*-rutinoside significantly decreased after 24 and 48 h, respectively, by 21.93% and 18.56%, but increased after 72 h by 6.29% compared with the control variant (Table 3).

The quantities of myricetin after 24 h of aerobic solid-phase fermentation did not change significantly, but after 48 and 72 h of the fermentation process, it increased, respectively, by 19.17% and 30.83% compared with non-fermented fireweed leaves. After 24 h of anaerobic solid-phase fermentation, the amount of myricetin decreased by 4.66%, but during the further process, the myricetin content significantly increased: after 48 h, by 9.59% and after 72 h, by 34.20%.

The amounts of flavonoid luteolin increased after 48 and 72 h of aerobic solid-phase fermentation, respectively, by 5.29% and 7.65%, but did not change after 24 h of the fermentation process under aerobic conditions, compared with the control variant. The same tendency was seen after anaerobic solid-phase fermentation: after 24 h, the amount of luteolin did not change, but after 48 and 72 h, it increased, respectively, by 4.12% and 8.82%, compared with non-fermented fireweed leaves.

The amounts of quercetin did not change significantly after 24 h of solid-phase fermentation under aerobic conditions, compared with non-fermented fireweed leaves, but increased after 48 h and 72 h of the fermentation process, respectively, by 17.04% and 14.81%. After anaerobic solid-phase fermentation, the content of quercetin significantly increased after 24 h, 48 h and 72 h, respectively: 3. 70%, 11.11% and 18.52%.

Quantities of quercetin-3-*O*-glucoside decreased, compared with the control variant, after 24, 48 and 72 h of aerobic solid-phase fermentation, respectively, by 22.79%, 23.45% and 16.47%. After 24 h of anaerobic solid-phase fermentation, the amount of quercetin-3-*O*-glucoside was lower, by 35.62%, compared with non-fermented fireweed leaves, but after 48 h and 72 h of the fermentation process, the content of quercetin-3-*O*-glucoside had an opposite tendency and significantly increased, respectively, by 10.62% and 172.02%.

The quantities of kaempferol increased in all aerobic solid-phase fermentation variants by 2.40% after 24 h, by 1.20% after 48 h and by 1.80% after 72 h, compared with non-fermented fireweed leaves. After anaerobic solid-phase fermentation, the tendency was similar (increased by 0.60% after 24 h, by 0.60% after 48 h and by 2.99% after 72 h) compared with non-fermented fireweed leaves (Table 3).

### 2.4. Antioxidant Activity

ABTS assay belongs to the most popular methods employed for estimating antioxidant activity. Thus, we have presented the antioxidant activity determined by an ABTS^•+^ method; besides, only this method was included in the methodology of this experiment.

We noticed that the antioxidant activity of fireweed leaves samples fermented under different fermentation methods and different fermentation duration significantly differed (Figure 1). The antioxidant activity after aerobic fermentation resulted in a significant increase from 5.64% to 17.63%. The greatest antioxidant activity (378.17 mmol TEAC 100 g^−1^ DW) in comparison with other investigated fireweed leaves samples has been determined after 72 h duration of anaerobic solid-phase fermentation. Under anaerobic conditions, only the long duration (72 h) gave higher results of the antioxidant activity of fireweed leaves compared with the control treatment (non-fermented leaves). The lowest antioxidant activity (230.77 mmol TEAC 100 g^−1^ DW) was determined in the leaves after 24 h of anaerobic solid-phase fermentation.

The present study determined that the antioxidant activity of bioactive compounds in fireweed leaves correlated (r = 0.7643, *p* = 0.00005) with the total amount of polyphenols (Figure 2). This demonstrates that with the increasing amount of polyphenols, the antioxidant activity increases as well.

### 2.5. Principal Component Analysis

Principal component analysis (PCA) was applied to establish the relationships among fireweed leaves samples and variables. The PCA biplot graphically summarized the allocation of data for the 15 bioactive substances (Table 1, Table 2 and Table 3) and antioxidant activity (Figure 1).

The results of PCA showed that the first two principal components (the first axis (PC1) and second axis (PC2)) described 67.37% of the total variance of bioactive substances in fireweed leaves samples (Figure 3). PC1 depicts 47.28% of the total variance, and PC2, 20.08% of the total variance. The eigenvalues of PC1 and PC2 were higher than one (7.56 and 3.21, respectively). As shown in Figure 3, the total polyphenols, *p*-coumaric acid, oenothein B, total flavonoids, quercetin-3-*O*-rutinoside, myricetin, luteolin, quercetin, quercetin-3-*O*-glucoside and antioxidant activity were highly positively associated with PC1. The PC2 was positively associated with total phenolic acids and ellagic acid.

Anaerobic SPF 72 h samples positively associated with PC1 are particularly characterized by high amounts of the total polyphenols, *p*-coumaric acid, oenothein B, total flavonoids, quercetin-3-*O*-rutinoside, myricetin, luteolin, quercetin, quercetin-3-*O*-glucoside and antioxidant activity. While Anaerobic SPF 24 h sample negatively associated on the first axis are defined by the low amount of previously mentioned components. NF (control) and Anaerobic SPF 48 h samples positively associated with PC2 are mainly characterized by high amounts of total phenolic acids and ellagic acid. However, the Aerobic SPF 48 h sample was negatively associated with PC2.

## 3. Discussion

According to Schepetkin et al. [29], oenothein B is a compound that corresponds to one of the major components of the biologically active extracts of fireweed leaves. Changes in oenothein B content during solid-phase fermentation are shown in Table 1. It was found that the technological process of solid-phase fermentation used in the experiment affected its content. The amounts of oenothein B depend on the amounts of ellagic and gallic acids. In addition, the crushing and pressing of the leaves during solid-phase fermentation can enhance the degradation processes of the cell walls and thus improve the diffusion of biologically active substances from the inner parts of the cell, which leads to more efficient extraction of compounds. It is believed that this may have been one of the main factors in determining higher oenothein B levels after the solid-phase fermentation process.

Gallic acid is identified in plants as part of hydrolyzable tannins or in free form. The groups of gallic acid and ellagic acid are structurally related. The results obtained in this experiment can be explained by the degradation of hydrolyzed tannins (gallotannins and ellagitannins) and ellagic acid to gallic acid and glucose during hydrolysis [30]. Cinnamic acid is involved in the metabolism of chlorogenic and *p*-coumaric acids [31], leading to qualitative and quantitative changes in these phenolic acids during solid-phase fermentation (Table 2).

During fermentation, ellagitannins are hydrolyzed to ellagic acids [32]. During solid-phase fermentation, the activity of microorganisms (such as lactic acid bacteria, yeast, etc.) and the activity of enzymes (polyphenolic oxidases) are activated, so the part of the ellagic acid decomposes into lower molecular weight acids (gallic acids). This fact can explain the reduced amounts of ellagic acid in almost all variants of fermented leaves (Table 2). A similar tendency was in another study, where the tannins were the most deeply affected compounds, with a 93% loss of content [24].

Benzoic acid is found in many plants. This acid mediates the biosynthesis of many secondary metabolites. In addition, bacterial metabolism and active enzyme activity determine the synthesis of benzoic acid from macromolecular compounds [33], which could explain the obtained experimental results (Table 2).

The flavonoids quercetin, quercetin-3-*O*-rutinoside (rutin) and quercetin-3-*O*-glucoside are biochemically and structurally highly related to each other. Myricetin, luteolin, and quercetin are also structurally similar, so myricetin can be synthesized from kaempferol [34]. These flavonoids’ biochemical, structural similarities and activity of microorganisms (such as lactic acid bacteria, etc.) in the solid-phase fermentation process activated the catabolic and anabolic processes of these biologically active substances in the fireweed leaves, which are thought to have led to the obtained results. According to Olennikov [24], the changes in the total phenolic content were not significant; however, inside this class of compounds, significant changes were observed.

Many medicinal plants which are rich in polyphenolic compounds have higher antioxidant effects. All parts of the fireweeds have strong antioxidant activity [35]. Many methods have been developed to evaluate antioxidant activity, which is based on the detection of free radicals that interact with antioxidants. Usually, DPPH and ABTS methods are used [36]. The ABTS radical-cation coupling method was used to evaluate the tendency of the antioxidant activity of the biologically active substances present in the leaves of fireweeds (Figure 1). The antioxidant activity of bioactive compounds in fireweed leaves positively correlated with the total amount of polyphenols: with the increased amount of polyphenols, the antioxidant activity also increased (Figure 2).

Summarizing the data obtained, it can be stated that the highest antioxidant activity in fireweed leaves was found after 72 h of the aerobic solid-phase fermentation process and the lowest in non-fermented fireweed leaves. But other scientists found that the antioxidant activity parameter of non-fermented leaves of *E. angustifolium* was not considerably different from that of fermented leaves, thus, indicating statistically insignificant differences in the antioxidant properties of the two plant materials [24].

Based on the obtained research data, it can be stated that different conditions of solid-phase fermentation (aerobic and anaerobic) and different duration of fermentation (24, 48 and 72 h) affected the changes in the amount of biologically active substances (flavonoids, phenolic acids, tannins) in the leaves of fireweeds. Scientists Couto and Sanroman [37] stated that the quality of the solid-phase fermentation process could be influenced not only by the duration of fermentation but also by other factors: certain microorganisms (lactic acid bacteria and others) and yeast [38].

## 4. Materials and Methods

### 4.1. Object of Research

The leaves of fireweed (*Chamerion angustifolium* (L.) Holub) for the experiment were collected from a naturally-growing fireweed habitat located in Giedres Nacevicienes ecological farm (No. SER-T-19-00910), which is in Safarka village (55°00′22″ N; 24°12′22″ E), Jonava district (Lithuania) in 2021. The total plot area of the experiment (natural habitat) was 12 ares (1200 m^2^).

### 4.2. Methods for Determination

The leaves of fireweeds were randomly collected from different parts of the experimental field at the stage of full flowering in early July (I decade). For the raw material collection studies, the experiment was performed in four replicates. The variants in the repetition were arranged randomized. Plant protection products against diseases and pests have not been used. In each replicate, 50 plants were selected for the study. 10 to 15 leaves were collected from each plant. The leaves were collected during the day at about noon when they were dry. The composite sample of the leaves was 6.3 kg and divided for further experiments:Control: 0.900 kg non-fermented fireweed leaves (0 h).Aerobic: 2.7 kg for solid-phase fermentation lasting 24, 48 and 72 h.Anaerobic: 2.7 kg for solid-phase fermentation lasting 24, 48 and 72 h (Table 4).

The process of solid-phase fermentation: the fresh leaves of fireweeds were carefully crushed and cut with special plastic knives. Samples of fireweed leaves were divided into 0.300 kg for each of the three replications. For anaerobic solid-phase fermentation, the leaves were rigidly crushed in glass containers (150 mL capacity) and covered with a lid, and for aerobic solid-phase fermentation, the leaves were rigidly crushed in glass containers (150 mL capacity) and covered with an air-passing lid. The solid-phase fermentation process took place at 24, 48 and 72 h in a controlled growth chamber (Sanyo, Growth Cabinet MLR-350, Moriguchi, Japan) (Laboratory of Agrobiotechnology, Agriculture Academy, Vytautas Magnus University) at a temperature of 30 ± 0.5 °C. Control and fermented leaf samples were dried at 40 °C for 10 h using a Termaks drying oven (Bergen, Norway). Then all samples were grounded with a powder grinder Grindomix GM 200 mill (Retsch GmbH, Haan, Germany) and stored in closed packages at 25 °C in a dry, dark, well-ventilated room. All biochemical fireweed leaves analyzes were performed in triplicate.

### 4.3. Methods of Laboratory Research and Analyses

The chemical composition and antioxidant activity of the leaves of fireweeds were analyzed in the Laboratory of Biochemical Research at the Warsaw University of Life Sciences (Warsaw, Poland). Separation and identification of flavonoids, tannins, phenolic acids and determination of antioxidant activity were performed in the dry matter (d.m.):

Polyphenolic compounds were quantitatively and qualitatively evaluated by high-performance liquid chromatography (HPLC) previously described by Ponder and Hallmann [39]. 100 mg of dried plant leaves powder was mixed with 5 mL of 80% methanol and shaken with a Micro-Shaker 326 (Marki, Poland). Then all samples were extracted in an ultrasonic bath (10 min, 30 °C, 5.5 kHz). After 15 min, the extraction samples were centrifuged (10 min, 3780× *g*, 5 °C). The supernatant was carefully collected in a clean plastic tube and centrifuged again (5 min, 31,180× *g*, 0 °C). A total of 850 μL of supernatant was transferred to the HPLC vial. A Synergi Fusion RP 80i Phenomenex column (250 × 4.60 mm) was used to separate and identify polyphenolic compounds. Analysis was performed using Shimadzu equipment (US Manufacturing Inc., Lebanon, IN, USA: two pumps LC-20AD, controller CBM-20A, column oven SIL-20AC, spectrometer UV/Vis SPD-20 AV). Polyphenolic compounds were separated under gradient conditions at a flow rate of 1 mL min^−1^. Two gradient phases were used: 10% (*V*:*V*) acetonitrile and ultrapure water (phase A); 55% (*V*:*V*) acetonitrile and ultrapure water (phase B). The phases were acidified with orthophosphoric acid (pH 3.0). The total analysis time was 38 min. The phase time program was as follows: 1.00 to 22.99 min, 95% for phase A and 5% for phase B; 23.00–27.99 min, 50% for phase A and 50% for phase B; 28.00–28.99 min, 80% for phase A and 20% for phase B; 29.00–38.00 min, 95% of phase A and 5% of phase B. A wavelength of 250 nm was chosen for flavonoids, 370 nm for phenolic acids and 570 nm for oenothein B. Polyphenolic compounds were identified using 99.90% purity standards (Sigma-Aldrich, Szelagowska, Poland) and standard analysis time.

Antioxidant activity was determined by the method previously described Srednick-Tober et al. [40]. 250 mg of dried fireweed leaves powder was weighed and poured into a plastic container and poured into 25 mL of distilled water. The samples were mixed in a mill (Labo Plus, Warsaw, Poland) for 1 min. The samples were placed in a mixer incubator (IKA, Staufen im Breisgau, Germany) for 1 h (at a temperature of 30 °C). Then the samples were shaken again and centrifuged (Centrifuge MPW-380, Warsaw, Poland) at 5 °C and 14,560× *g* for 15 min. In the next step, the supernatant was collected for identification. The solutions of the leaf extracts in the laboratory glassware were measured according to the prescribed dilution scheme (0.5–1.5 mL), and then up to 3 mL of ABTS^•+^ (2,2′-Azino-bis (3-ethylbenzothiazoline-6-sulfonic acid) cationic solution with PBS (phosphate buffered saline) was added. After 6 min, the absorbances of the samples were determined (21 °C, wavelength λ = 734 nm) using a spectrophotometer (Helios γ, Thermo Scientific, Warsaw, Poland) according to a special formula, including the dilution factor. The final results were expressed in mmol TE (Trolox equivalent/100 g dry matter).

### 4.4. Mathematical Statistical Evaluation of Research Data

All data were statistically processed using analysis of variance (ANOVA) from the STATISTICS software package (Statistica 10; StatSoft, Inc., Tulsa, OK, USA). The statistical significance of differences between means was assessed by Fisher’s LSD test (*p* < 0.05). Correlation regression analysis was performed to determine the nature and strength of the relationships between the variables. To assess the relationships between different fermentation conditions (aerobic and anaerobic environment, duration) and changes in the amounts of biologically active substances (flavonoids, tannins, phenolic acids), principal component analysis (PCA) was performed.

## 5. Conclusions

The discussed results showed that solid-phase fermentation affected the amounts of biologically active substances in the leaves of fireweeds. The highest content of oenothein B was after 72 h of solid-phase fermentation process under anaerobic conditions, so it would be appropriate to recommend these fermented fireweed leaves as a plant source of tannin oenothein B, which has anticancer properties.

Total polyphenols content increased significantly after 72 h under both—aerobic and anaerobic solid-phase fermentation, total flavonoids also increased after 72 h of anaerobic fermentation, but the total phenolic acid content decreased after long fermentation, compared with the control. Therefore, according to these data, it would be possible to recommend fireweed leaves after 72 h of anaerobic fermentation as a source of flavonoids (especially quercetin-3-*O*-glucoside and quercetin-3-*O*-rutinoside) and 48 h anaerobic fermentation or non-fermented leaves as a source of ellagic acid, which is the most abundant phenolic acid in fireweeds and has anticancer properties.

In summary, we can say that solid-phase fermentation activates both the degradation of cell walls and the activity of microorganisms and enzymes, which leads to quantitative and qualitative changes in biologically active substances. Based on the data of this study, 72 h anaerobic solid-phase fermentation could be suggested for the food and pharmacy industry to produce fireweed leaves functional food products with higher flavonoids and tannin oenothein B, and non-fermented leaves of fireweeds could be very useful for people due to high phenolic acids content. Also, based on these results, the conditions for solid-phase fermentation can be further improved.

## Figures and Tables

**Figure 1 plants-12-00277-f001:**
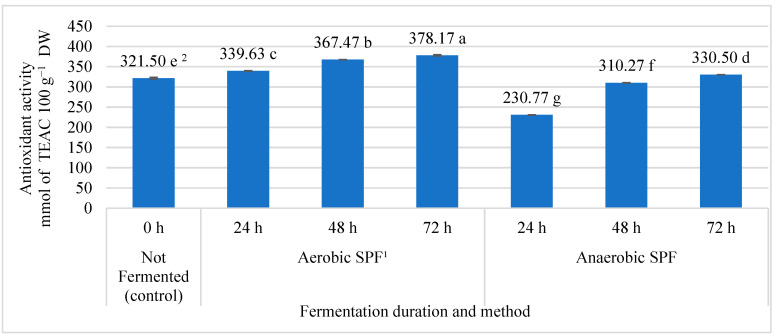
The influence of solid-phase fermentation on antioxidant activity in fireweed leaves. ^1^ SPF—solid-phase fermentation. ^2^ Averages marked by different small letters are statistically significantly different at *p* < 0.05.

**Figure 2 plants-12-00277-f002:**
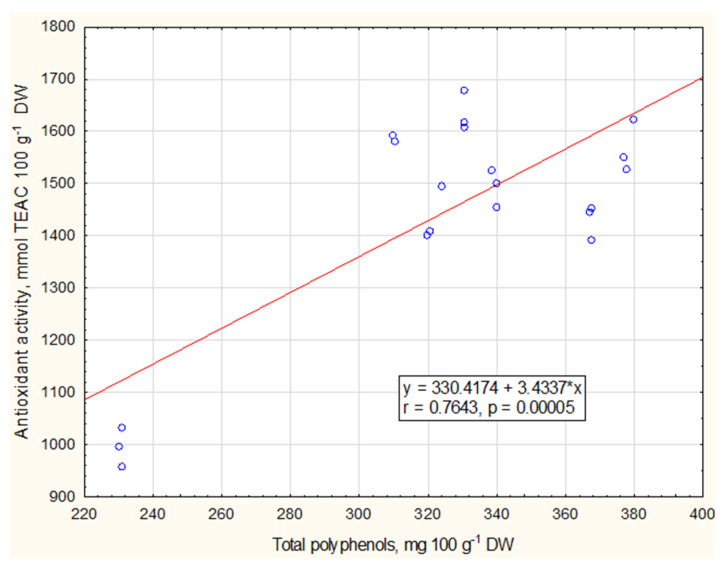
The correlation between total polyphenols and antioxidant activity in fireweed leaves.

**Figure 3 plants-12-00277-f003:**
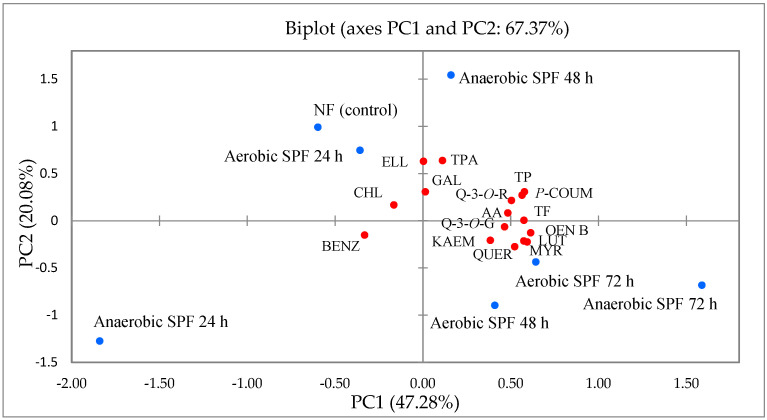
Principal component analysis of flavonoids, phenolic acids, oenothein B (tannin) and antioxidant activity in fireweed leaves samples fermented under the different fermentation methods and different fermentation duration, based on 16 variables. Variables: total polyphenols (TP), total phenolic acids (TPA), ellagic acid (ELL), gallic acid (GAL), chlorogenic acid (CHL), *p*-coumaric acid (*P*-COUM), benzoic acid (BENZ), total flavonoids (TF), quercetin-3-*O*-rutinoside (Q-3-*O*-R), myricetin (MYR), luteolin (LUT), quercetin (QUER), quercetin-3-*O*-glucoside (Q-3-*O*-G), kaempferol (KAEM), oenothein B (OEN B) and antioxidant activity (AA). Conditions of fermentation: fermentation method (aerobic solid-phase fermentation method (Aerobic SPF), anaerobic solid-phase fermentation method (Anaerobic SPF), not fermented (NF)) and their different duration (Not fermented (NF), fermented 24 h, 48 h and 72 h).

**Table 1 plants-12-00277-t001:** The influence of solid-phase fermentation on the contents of total polyphenols, total phenolic acids, total flavonoids and Oenothein B (tannin) in fireweed leaves.

Bioactive Group/ Fermentation Duration	Total Polyphenols	Total Phenolic Acids	Total Flavonoids	Oenothein B (Tannin)
	mg 100 g^−1^ Dry Weight (DW)			
Not fermented (control)	1434.02 ± 51.96 b ^1^	845.52 ± 35.83 b	38.69 ± 0.50 b	549.81 ± 18.08 d
	Aerobic solid-phase fermentation method			
Fermented 24 h	1493.37 ± 35.61 b	842.91 ± 28.01 b	35.96 ± 1.58 c	614.50 ± 7.73 c
Fermented 48 h	1428.61 ± 33.37 b	631.57 ± 20.31 d	36.91 ± 0.79 c	760.13 ± 49.03 b
Fermented 72 h	1566.86 ± 49.63 a	729.96 ± 5.03 c	38.67 ± 0.53 b	798.22 ± 52.66 ab
	Anaerobic solid-phase fermentation method			
Fermented 24 h	994.79 ± 37.39 c	600.55 ± 31.39 d	20.69 ± 0.11 d	373.55 ± 15.65 e
Fermented 48 h	1584.30 ± 6.88 a	991.35 ± 13.01 a	38.61 ± 0.31 b	554.24 ± 10.64 d
Fermented 72 h	1633.96 ± 38.74 a	740.89 ± 9.57 c	70.13 ± 0.42 a	822.94 ± 34.54 a
*p*-Value (SPF ^2^ duration × SPF method)	<0.00001	<0.00001	<0.00001	<0.00001

^1^ Averages in the same columns marked by different small letters are statistically significantly different at *p* < 0.05. ^2^ SPF—solid-phase fermentation.

**Table 2 plants-12-00277-t002:** The influence of solid-phase fermentation on the contents of individual phenolic acids in fireweed leaves.

Bioactive Group/ Fermentation Duration	Ellagic	*P*-Coumaric	Gallic	Chlorogenic	Benzoic
	mg 100 g^−1^ Dry Weight (DW)				
Not fermented (control)	534.15 ± 22.49 b ^1^	213.81 ± 9.52 c	29.14 ± 2.18 e	59.08 ± 6.84 a	9.33 ± 0.39 b
	Aerobic solid-phase fermentation method				
Fermented 24 h	464.75 ± 17.16 c	202.11 ± 9.94 cd	135.20 ± 1.72 a	28.92 ± 0.57 c	11.93 ± 0.63 a
Fermented 48 h	334.91 ± 12.78 e	197.64 ± 9.97 d	66.57 ± 1.43 c	27.02 ± 0.78 c	5.42 ± 0.11 d
Fermented 72 h	406.08 ± 5.88 d	211.20 ± 4.92 cd	45.11 ± 1.06 d	58.47 ± 4.13 a	9.10 ± 1.24 bc
	Anaerobic solid-phase fermentation method				
Fermented 24 h	354.83 ± 26.55 e	156.13 ± 8.41 e	38.10 ± 6.08 d	40.03 ± 4.66 b	11.46 ± 0.06 a
Fermented 48 h	622.15 ± 11.53 a	234.11 ± 6.12 b	91.14 ± 9.61 b	39.73 ± 5.42 b	4.23 ± 0.71 e
Fermented 72 h	406.62 ± 5.27 d	255.73 ± 10.94 a	44.22 ± 0.81 d	26.22 ± 0.61 c	8.11 ± 0.32 c
*p*-Value (SPF ^2^ duration × SPF method)	<0.00001	<0.00001	<0.00001	<0.00001	<0.59724

^1^ Averages in the same columns marked by different small letters are statistically significantly different at *p* < 0.05. ^2^ SPF—solid-phase fermentation.

**Table 3 plants-12-00277-t003:** The influence of solid-phase fermentation on the contents of individual flavonoids in fireweed leaves.

Bioactive Group/ Fermentation Duration	Quercetin-3-*O*-Rutinoside	Myricetin	Luteolin	Quercetin	Quercetin-3-*O*-Glucoside	Kaempferol
	mg 100 g^−1^ Dry Weight (DW)					
Not fermented (control)	13.36 ± 0.30 b ^1^	3.86 ± 0.09 d	1.70 ± 0.01 d	1.35 ± 0.01 e	16.76 ± 0.13 c	1.67 ± 0.007 e
	Aerobic solid-phase fermentation method					
Fermented 24 h	14.51 ± 0.19 a	3.76 ± 0.05 de	1.69 ± 0.01 d	1.34 ± 0.01 e	12.94 ± 1.44 d	1.71 ± 0.003 a
Fermented 48 h	14.41 ± 0.34 a	4.60 ± 0.03 b	1.79 ± 0.01 c	1.58 ± 0.01 a	12.83 ± 0.48 d	1.69 ± 0.002 c
Fermented 72 h	14.54 ± 0.15 a	5.05 ± 0.08 a	1.83 ± 0.01 b	1.55 ± 0.02 b	14.00 ± 0.31 d	1.70 ± 0.001 b
	Anaerobic solid-phase fermentation method					
Fermented 24 h	10.43 ± 0.03 d	3.68 ± 0.03 e	1.70 ± 0.01 d	1.40 ± 0.01 d	10.79 ± 0.15 e	1.68 ± 0.001 d
Fermented 48 h	10.88 ± 0.35 c	4.23 ± 0.15 c	1.77 ± 0.01 c	1.50 ± 0.01 c	18.54 ± 0.11 b	1.68 ± 0.002 d
Fermented 72 h	14.20 ± 0.59 a	5.18 ± 0.07 a	1.85 ± 0.02 a	1.60 ± 0.02 a	45.59 ± 0.92 a	1.72 ± 0.003 a
*p*-Value (SPF ^2^ duration × SPF method)	<0.00001	<0.00048	<0.02033	<0.00001	<0.00001	<0.00001

^1^ Averages in the same columns marked by different small letters are statistically significantly different at *p* < 0.05. ^2^ SPF–solid-phase fermentation.

**Table 4 plants-12-00277-t004:** Laboratory experiment variants.

Fermentation Duration			
0 h	24 h	48 h	72 h
0.900 kg control (not fermented)	0.900 kg aerobic SPF ^1^	0.900 kg aerobic SPF	0.900 kg aerobic SPF
	0.900 kg anaerobic SPF	0.900 kg anaerobic SPF	0.900 kg anaerobic SPF

^1^ SPF—solid-phase fermentation.

## Data Availability

Not applicable.

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
