# Peer review of "The Impact of Solid-Phase Fermentation on Flavonoids, Phenolic Acids, Tannins and Antioxidant Activity in *Chamerion angustifolium* (L.) Holub (Fireweed) Leaves"

_plants, 2023, doi:10.3390/plants12020277_

Round 1
Reviewer 1 Report
Attach fie

Reviewer 2 Report
In introduction, each paragraph starts with fireweeds. Additionally, sentence after sentence starts with fireweeds. There are information’s that are not important.
Meaningful discussion is needed; why there is lower amount of polyphenols after 48 h fermentation and 72 while in anaerobic condition there is different trend. This is for total amount as well as for some individual compound. Fluctuation of results is quite questionable. Why gallic acid is in such a high amount after 24 h fermentation; all other samples have significantly lower amount of this acid – this should be check with additional studies if it is correct. Discussion have to be improved; comparison with other studies which used solid phase fermentation if they obtained such fluctuation of results, since it might be a problem with starting sample.
Only one assay was used for evaluation of AA; other methods have to be applied in order to obtain better picture of AA of samples.
r = 0.7643 is not strong correlation
Organization of material and methods section have to be improved.
English has to be checked and improved.
Reviewer 3 Report
Dear Author, I reviewed the manuscript (plants-2079569) entitled The Impact of Solid-phase Fermentation on Flavonoids, Phenolic acids, Tannins and Antioxidant Activity in Chamerion angustifolium (L.) Holub (fireweed) Leaves. This manuscript presents relevant information about solid-phase fermentation's impact on antioxidant potential of Chamerioun angustifolium. However, some sections of the presented data can be improved. For this reason, I consider that this manuscript needs minor changes to be considered for publication in this journal.
Additional comments.
Highlight the advantages of using Chamerion angustifolium as antioxidant source.
Check the paragraph extension in this manuscript.
Include an experimental design containing statistical factors and response variables in the statistical analyses applied to the findings of this research.
Try to discuss the obtained findings with similar works where similar plants were processed under solid-phase fermentation and antioxidant properties were evaluated.
Include future trends to keep working with the obtained data.
Try to conclude with a general statement of the most relevant part of this study.
Round 2
Reviewer 2 Report
It is not possible to understend figure 2.
